



# A new vertically integrated, MOno-Layer Higher-Order ice flow model (MOLHO)

Thiago Dias dos Santos[1,2], Mathieu Morlighem[1,3], and Douglas Brinkerhoff[4]

[1]Department of Earth System Science, University of California, Irvine, CA, USA
[2]Centro Polar e Climático, Universidade Federal do Rio Grande do Sul, Porto Alegre, RS, Brazil
[3]Department of Earth Sciences, Dartmouth College, Hanover, NH, USA
[4]Department of Computer Science, University of Montana, Missoula, MT, USA

**Correspondence:** Thiago Dias dos Santos (santos.td@gmail.com)

**Abstract.** Numerical simulations of ice sheets rely on the momentum balance to determine how ice velocities change as the geometry of the system evolves. Ice is generally assumed to follow a Stokes flow with a nonlinear viscosity. Several approximations have been proposed in order to lower the computational cost of a full-Stokes stress balance. A popular option is the Blatter-Pattyn or Higher-Order model (HO), which consists of a three-dimensional set of equations that solves the horizontal velocities only. However, it still remains computationally expensive for long transient simulations. Here we present a depth-integrated formulation of the HO model, which can be solved on a two-dimensional mesh in the horizontal plane. We employ a specific polynomial function to describe the vertical variation of the velocity, which allows us to integrate the vertical dimension using a semi-analytic integration. We assess the performance of this MOno-Layer Higher-Order model (MOLHO) to compute ice velocities and simulate grounding line dynamics on standard benchmarks (ISMIP-HOM and MISMIP3D). We compare MOLHO results to the ones obtained with the original three-dimensional HO model. We also compare the time performance of both models in time-dependent runs. Our results show that the ice velocities and grounding line positions obtained with MOLHO are in very good agreement with the ones from HO. In terms of computing time, MOLHO requires less than 10% of the computational time of a typical HO model, for the same simulations. These results suggest that the MOno-Layer Higher-Order formulation provides improved computational time performance and a comparable accuracy compared to the HO formulation, which opens the door to Higher-Order paleo simulations.

## 1 Introduction

Projecting the future evolution of the ice sheets of Greenland and Antarctica and their potential contribution to sea level rise often relies on computer simulations carried out by numerical ice sheet models (e.g. Aschwanden et al., 2019; Goelzer et al., 2020; Seroussi et al., 2020; Edwards et al., 2021). These ice sheet models solve a set of flow equations based on the conservation of momentum to obtain the ice velocity field over the entire ice sheet. There exist several ice flow models in the field of ice sheet/glacier modeling. One of these flow models is the Full-Stokes equations (FS), a set of equations that solve the three-dimensional ice velocity as well as the ice pressure. The FS model is applicable to a wide range of systems: from





small glaciers to continental-sized ice sheets. However, this model is computationally demanding, especially in time-dependent numerical simulations, which restricts its use to short term projections or regional applications.

Over the last decades, simplified ice flow models were developed in order to decrease the computational cost of the momentum balance (e.g, Hutter, 1983; Morland, 1987; MacAyeal, 1989; Blatter, 1995; Pattyn, 2003; Goldberg, 2011). All of them are derived from the FS equations and rely on different approximations in the stress tensor or velocity field. A commonly applied approximation is the Blatter-Pattyn model (Blatter, 1995; Pattyn, 2003), also known as a Higher-Order model (HO). The HO model assumes a hydrostatic pressure and negligible contribution of horizontal gradient of vertical velocities to the

velocity computation. The model is three-dimensional and solves the horizontal velocities over the entire ice sheet. However, for long transient runs and/or sensitivity analyses that require to run the model for large ensembles, the HO model still demands relatively high computational resources. Recently, numerical schemes were applied to improve the computational efficiency of the HO model without compromising numerical accuracy (e.g., Bassis, 2010; Brinkerhoff and Johnson, 2015; Cuzzone et al., 2018; Shapero et al., 2021). These schemes rely on the finite element method (FEM), which allows to employ polynomial

functions of degree equal or greater than 2 to model the vertical variation of the horizontal velocities. These approaches decrease the number of vertical elements (layers) in the mesh, reducing the overall computational cost. A natural extension of such schemes is to employ a single layer of three-dimensional elements (e.g., triangular prisms) with a higher order polynomial along the vertical axis (Brinkerhoff and Johnson, 2015; Shapero et al., 2021). This mono-layer, higher-order flow model presents a reasonable numerical accuracy in comparison to the original three-dimensional HO model, at a relatively lower com-

putational cost (Brinkerhoff and Johnson, 2015). It has been used for simulations of glaciers and ice sheets (Brinkerhoff and Johnson, 2015; Brinkerhoff et al., 2017; Shapero et al., 2021). However, some technical aspects, such as the vertical integration of the viscosity, have not been fully resolved and the performance of this mono-layer higher-order model in marine ice sheet simulations has not been tested.

    Here we present a finite element formulation of a new vertically integrated, MOno-Layer Higher-Order (MOLHO) ice

flow model inspired by the scheme employed by Brinkerhoff and Johnson (2015). Previous works have employed numerical integration (e.g., Gauss-Legendre quadrature) over the vertical dimensional in triangles (Brinkerhoff and Johnson, 2015) or triangular prisms (Shapero et al., 2021) to account for the vertical shear and ice viscosity. Here, we pre-compute a depth-average the ice viscosity, which allows to perform an analytical vertical integration. Thus, the new formulation is written over the horizontal plane and relies on a two-dimensional mesh. The vertical variation of the horizontal velocities is described by a

specific higher-order polynomial. This specific polynomial adds two degrees of freedom per node and allows us to integrate over the vertical axis. The vertical integration of the ice viscosity is evaluated by a semi-analytic scheme of low computational cost. The formulation is implemented in the Ice-sheet and Sea-level System Model v4.18. We first run the ISMIP-HOM diagnostic experiments (Pattyn et al., 2008) to evaluate the velocity field for different wavelengths of bedrock elevation and compare the results to the solutions obtained using the three-dimensional HO model. We then run the MISMIP3D benchmark (Pattyn et al.,

2013) to evaluate the behavior of grounding line dynamics in a marine ice sheet setup. With the MISMIP3D experiments we assess the grounding line position at steady state for different mesh resolutions as well as grounding line reversibility after





imposed friction perturbations. We also run a time performance analysis to compare the simulation time spent by each flow model, MOLHO and HO, and conclude on the applicability of this model in the future.

The paper is organized as follows. We first present the description of the MOLHO formulation (Sect. 2), highlighting the main steps necessary to perform the vertical integration of the weak formulation (e.g, Sect. 2.3), especially the integration of the ice viscosity (Sect. 2.4). Then, we describe the numerical experiments performed to evaluate the new formulation (Sect. 3), followed by the results (Sect. 4). We finish the paper with a discussion (Sect. 5) and a summary of the conclusions of this work (Sect. 6). Additional information regarding the pre-computation of the ice viscosity as well as the resulting element stiffness matrix and load vector is provided in the Appendix sections.

## 2 Description of the MOno-Layer Higher-Order (MOLHO) model

### 2.1 Polynomial function for vertical shear

The specific polynomial employed to describe the vertical variation of the horizontal velocities is based on the Shallow Ice Approximation (SIA, Hutter, 1983). In SIA, the horizontal velocities ($\boldsymbol{v} = [v_x, v_y]^{\mathsf{T}}$) of an isothermal ice sheet following Glen's law are given by:

$$\boldsymbol{v} = \boldsymbol{v}^b - 2\left(\rho g\right)^n \left\|\nabla s\right\|^{n-1} \frac{A}{n+1} H^{n+1} \left(1 - \left(\frac{s-z}{H}\right)^{n+1}\right) \nabla s, \tag{1}$$

where $\boldsymbol{v}^b \left(= \left[v_x^b, v_y^b\right]^{\mathsf{T}}\right)$ are the basal horizontal velocities, $H$ is the ice thickness, $s$ is the ice surface elevation, $A$ is the ice rate factor, $n$ is the Glen's law exponent, $\rho$ is the ice density, and $g$ is the gravitational acceleration. We write Equation 1 as:

$$\boldsymbol{v}\left(x, y, z\right) = \boldsymbol{v}^b\left(x, y\right) + \boldsymbol{v}^{sh}\left(x, y\right)\left(1 - \zeta^{n+1}\right), \tag{2}$$

where $\boldsymbol{v}^{sh} = \left[v_x^{sh}, v_y^{sh}\right]^{\mathsf{T}}$ is the internal deformation contribution to the surface velocities, and $\zeta$ is an auxiliary variable defined as:

$$\zeta = \frac{s-z}{H} = 1 - \frac{z-b}{H}. \tag{3}$$

Equation 2 has been widely employed by the glaciological community in ice sheet models to setup boundary conditions (e.g., Raymond, 1983) and describe vertical deformation within the glacier body (e.g., Bueler and Brown, 2009; Bassis, 2010; Brinkerhoff and Johnson, 2015; Shapero et al., 2021).

In SIA, $\boldsymbol{v}^{sh}$ are defined using the local surface gradient only (Eq. 1). In the MOno Layer Higher-Order Model (MOLHO), we employ Eq. 2 into a Higher-Order (HO) weak formulation such that $\boldsymbol{v}^{sh}$ are evaluated only after solving a nonlinear system resulting from the HO stress-balance equations. Details of the computation of $\boldsymbol{v}^{sh}$ are shown in Sect. 2.3.

To derive the MOLHO formulation, we use some identities related to vertical derivatives and vertical integration of the function $1 - \zeta^{n+1}$. Appendix B presents some useful identities employed in this work.





## 2.2 Weak formulation

The three-dimensional Blatter-Pattyn or Higher-Order (HO) ice flow model is defined as (Blatter, 1995; Pattyn, 2003):

$$\frac{\partial}{\partial x}\left(4\mu\frac{\partial v_x}{\partial x}+2\mu\frac{\partial v_y}{\partial y}\right)+\frac{\partial}{\partial y}\left(\mu\left(\frac{\partial v_x}{\partial y}+\frac{\partial v_y}{\partial x}\right)\right)+\frac{\partial}{\partial z}\left(\mu\frac{\partial v_x}{\partial z}\right)=\rho g\frac{\partial s}{\partial x},$$

$$\frac{\partial}{\partial x}\left(\mu\left(\frac{\partial v_x}{\partial y}+\frac{\partial v_y}{\partial x}\right)\right)+\frac{\partial}{\partial y}\left(4\mu\frac{\partial v_y}{\partial y}+2\mu\frac{\partial v_x}{\partial x}\right)+\frac{\partial}{\partial z}\left(\mu\frac{\partial v_y}{\partial z}\right)=\rho g\frac{\partial s}{\partial y}. \tag{4}$$

For the boundary conditions, we assume a negligible stress at the ice surface (i.e., $\boldsymbol{\tau}^s = \mathbf{0}$) and a viscous friction law at the ice base defined as:

$$\boldsymbol{\tau}^b = -\alpha^2 \boldsymbol{v}^b, \tag{5}$$

where $\boldsymbol{v}^b = \left[v_x^b, v_y^b\right]^{\mathsf{T}}$ is the horizontal velocity at the glacier base and $\alpha = \alpha\left(x, y\right)$ is the friction coefficient that, in general, is a (nonlinear) function of the basal velocities $\boldsymbol{v}^b$. Note that other friction laws could be employed in this formulation. At the ice-ocean interface (i.e., calving front), a Neumann boundary condition based on ocean water pressure is applied.

Let $\mathcal{V}$ be the space of kinematically admissible fields that satisfy the Dirichlet boundary conditions and whose first derivatives are square integrable on the glacier domain. The weak formulation (assuming non-homogeneous Dirichlet conditions on the model boundary and viscous sliding at the base) of HO is:

$$\int_\Omega \left(4\mu\frac{\partial v_x}{\partial x}+2\mu\frac{\partial v_y}{\partial y}\right)\frac{\partial \vartheta_x}{\partial x}+\mu\left(\frac{\partial v_x}{\partial y}+\frac{\partial v_y}{\partial x}\right)\frac{\partial \vartheta_x}{\partial y}+\mu\frac{\partial v_x}{\partial z}\frac{\partial \vartheta_x}{\partial z}\,d\Omega$$

$$+\int_{\Gamma^b}\alpha^2 v_x^b \vartheta_x\,d\Gamma^b$$

$$=-\int_\Omega \rho g\frac{\partial s}{\partial x}\vartheta_x\,d\Omega+\int_{\Gamma^w}\left[\left(4\mu\frac{\partial v_x}{\partial x}+2\mu\frac{\partial v_y}{\partial y}\right)n_x+\mu\left(\frac{\partial v_x}{\partial y}+\frac{\partial v_y}{\partial x}\right)n_y\right]\vartheta_x\,d\Gamma^w,$$

$$\int_\Omega \mu\left(\frac{\partial v_x}{\partial y}+\frac{\partial v_y}{\partial x}\right)\frac{\partial \vartheta_y}{\partial x}+\left(4\mu\frac{\partial v_y}{\partial y}+2\mu\frac{\partial v_x}{\partial x}\right)\frac{\partial \vartheta_y}{\partial y}+\mu\frac{\partial v_y}{\partial z}\frac{\partial \vartheta_y}{\partial z}\,d\Omega$$

$$+\int_{\Gamma^b}\alpha^2 v_y^b \vartheta_y\,d\Gamma^b$$

$$=-\int_\Omega \rho g\frac{\partial s}{\partial y}\vartheta_y\,d\Omega+\int_{\Gamma^w}\left[\left(4\mu\frac{\partial v_y}{\partial y}+2\mu\frac{\partial v_x}{\partial x}\right)n_y+\mu\left(\frac{\partial v_x}{\partial y}+\frac{\partial v_y}{\partial x}\right)n_x\right]\vartheta_y\,d\Gamma^w,$$

$$v_x = v_x^D, \, v_y = v_y^D \text{ on } \Gamma^D,$$

$$\forall\,(\vartheta_x, \vartheta_y) \in \mathcal{V}, \tag{6}$$





where $\Omega$ is the three-dimensional domain of the glacier, $\Gamma^b$ is the ice base, $\Gamma^w$ is the vertical face of the glacier boundary (e.g., calving front), and $\Gamma^D$ is the glacier boundary where Dirichlet boundary conditions with values $v_x^D$, $v_y^D$ are imposed.

## 2.3 Finite element discretization

Based on SIA, we decompose the horizontal velocities in MOLHO following Eq. 2:

$$
\begin{aligned}
v_x\left(x,y,z\right) &= v_x^b\left(x,y\right) + v_x^{sh}\left(x,y\right)\left(1 - \zeta^{n+1}\right), \\
v_y\left(x,y,z\right) &= v_y^b\left(x,y\right) + v_y^{sh}\left(x,y\right)\left(1 - \zeta^{n+1}\right).
\end{aligned}
\tag{7}
$$

We note that the decomposition employed here differs from the approach adopted in previous work (Brinkerhoff and Johnson, 2015). In the latter, the horizontal velocities were parameterized as the sum of depth-averaged velocities and the deviation from it (rather than basal and shear velocities) such that the first component of the stress balance solution (i.e., depth-averaged velocities) could be used directly into the continuity equation that controls the advection of the ice thickness. Here, once $\boldsymbol{v}^b$ and $\boldsymbol{v}^{sh}$ are computed, the vertically-averaged velocities, $\bar{\boldsymbol{v}}$, are obtained by:

$$
\bar{\boldsymbol{v}}\left(x,y\right) = \boldsymbol{v}^b\left(x,y\right) + \boldsymbol{v}^{sh}\left(x,y\right)\frac{(n+1)}{(n+2)}.
\tag{8}
$$

To apply the finite element discretization to Eq. 6, we approximate the horizontal velocities by employing two-dimensional basis functions $\phi_j$ as follows:

$$
\begin{aligned}
v_x\left(x,y,z\right) &\approx \sum_{j=1}^{n_f} v_{x,j}^b \phi_j\left(x,y\right) + \sum_{j=1}^{n_f} v_{x,j}^{sh}\phi_j\left(x,y\right)\left(1 - \zeta^{n+1}\right), \\
v_y\left(x,y,z\right) &\approx \sum_{j=1}^{n_f} v_{y,j}^b \phi_j\left(x,y\right) + \sum_{j=1}^{n_f} v_{y,j}^{sh}\phi_j\left(x,y\right)\left(1 - \zeta^{n+1}\right),
\end{aligned}
\tag{9}
$$

with $v_{x,j}^b$, $v_{y,j}^b$ and $v_{x,j}^{sh}$, $v_{y,j}^{sh}$ being the values of the base velocities and surface shear velocities evaluated at nodal points $j$. In Eq. 9, $n_f$ is the total number of basis functions in the entire mesh. By decomposing the velocities into two independent components, we can define the basis functions on a horizontal two-dimensional mesh only, i.e., $\phi = \phi\left(x,y\right)$.

We choose a similar decomposition for the test functions $\vartheta_x$ and $\vartheta_y$:

$$
\begin{aligned}
\vartheta_x &= \vartheta_x^b + \vartheta_x^{sh}\left(1 - \zeta^{n+1}\right), \\
\vartheta_y &= \vartheta_y^b + \vartheta_y^{sh}\left(1 - \zeta^{n+1}\right),
\end{aligned}
\tag{10}
$$

and they are defined using the same basis functions $\phi_i$, as follows:

$$
\begin{aligned}
\vartheta_x\left(x,y,z\right) &= \sum_{i=1}^{n_f} \vartheta_{x,i}^b \phi_i\left(x,y\right) + \sum_{i=1}^{n_f} \vartheta_{x,i}^{sh}\phi_i\left(x,y\right)\left(1 - \zeta^{n+1}\right), \\
\vartheta_y\left(x,y,z\right) &= \sum_{i=1}^{n_f} \vartheta_{y,i}^b \phi_i\left(x,y\right) + \sum_{i=1}^{n_f} \vartheta_{y,i}^{sh}\phi_i\left(x,y\right)\left(1 - \zeta^{n+1}\right).
\end{aligned}
\tag{11}
$$



where $\vartheta_{x,i}^b$, $\vartheta_{y,i}^b$ and $\vartheta_{x,i}^{sh}$, $\vartheta_{y,i}^{sh}$ are arbitrary coefficients.

We insert Eq. 9 and Eq. 11 into the weak formulation (Eq. 6), and then we replace the ice viscosity $\mu$ by an *ad hoc* depth-averaged viscosity $\bar{\mu}$ (discussed in Sect. 2.4). Thus, we analytically integrate the resulting formulation along the vertical axis. This generates a set of equations defined over the horizontal $xy$-plane (two-dimensional mesh). The resulting nonlinear element matrix system is:

$$
\begin{bmatrix}
\mathbf{K}_{11} & \mathbf{K}_{12} & \mathbf{K}_{13} & \mathbf{K}_{14} \\
\mathbf{K}_{21} & \mathbf{K}_{22} & \mathbf{K}_{23} & \mathbf{K}_{24} \\
\mathbf{K}_{31} & \mathbf{K}_{32} & \mathbf{K}_{33} & \mathbf{K}_{34} \\
\mathbf{K}_{41} & \mathbf{K}_{42} & \mathbf{K}_{43} & \mathbf{K}_{44}
\end{bmatrix}
\begin{bmatrix}
\boldsymbol{v}_x^b \\
\boldsymbol{v}_x^{sh} \\
\boldsymbol{v}_y^b \\
\boldsymbol{v}_y^{sh}
\end{bmatrix}
=
\begin{bmatrix}
\boldsymbol{F}_1 \\
\boldsymbol{F}_2 \\
\boldsymbol{F}_3 \\
\boldsymbol{F}_4
\end{bmatrix},
\tag{12}
$$

where $\mathbf{K}_{ij}$ are matrices of size $n_{f,e} \times n_{f,e}$, with $n_{f,e}$ being the total number of basis functions defined on each element. Here, we employ triangular elements and P1-Lagrangian basis functions and, therefore, $n_{f,e} = 3$. In the element matrix system (Eq. 12), $\boldsymbol{v}_x^b$, $\boldsymbol{v}_x^{sh}$, $\boldsymbol{v}_y^b$, and $\boldsymbol{v}_y^{sh}$ are vectors evaluated on each element's node (e.g., $\boldsymbol{v}_x^b = \left[v_{x,1}^b, v_{x,2}^b, v_{x,3}^b\right]^{\mathsf{T}}$). The loading vectors $\boldsymbol{F}_i$ are also evaluated on all nodes of each element. Appendix D presents the detailed technical steps of the vertical integration and the resulting expressions of each term ($\mathbf{K}_{ij}$ and $\boldsymbol{F}_i$) of Eq. 12. We note that we still rely on a numerical integration over the horizontal plane to evaluate the element matrices and loading vectors of Eq. 12.

## 2.4    Vertically integrated ice viscosity

A delicate aspect of the analytical depth-integration of the HO weak formulation (Eq. 6) is how to handle the ice viscosity that is a function of ice velocity. The ice viscosity $\mu = \mu(x, y, z)$ is assumed to follow Glen's flow law (Glen, 1955):

$$
\mu = \frac{1}{2} \frac{B}{\dot{\varepsilon}_e^{(n-1)/n}},
\tag{13}
$$

where $B = A^{-1/n}$ is the associated rate factor, and $\dot{\varepsilon}_e$ is the effective strain rate tensor, which for HO is defined as:

$$
\dot{\varepsilon}_e = \sqrt{\dot{\varepsilon}_{xx}^2 + \dot{\varepsilon}_{yy}^2 + \dot{\varepsilon}_{xy}^2 + \dot{\varepsilon}_{xz}^2 + \dot{\varepsilon}_{yz}^2 + \dot{\varepsilon}_{xx}\dot{\varepsilon}_{yy}},
\tag{14}
$$

with each component defined as:

$$
\begin{aligned}
\dot{\varepsilon}_{xx} &= \frac{\partial v_x}{\partial x}, \\
\dot{\varepsilon}_{yy} &= \frac{\partial v_y}{\partial y}, \\
\dot{\varepsilon}_{xy} &= \frac{1}{2}\left(\frac{\partial v_x}{\partial y} + \frac{\partial v_y}{\partial x}\right), \\
\dot{\varepsilon}_{xz} &= \frac{1}{2}\frac{\partial v_x}{\partial z}, \\
\dot{\varepsilon}_{yz} &= \frac{1}{2}\frac{\partial v_y}{\partial z}.
\end{aligned}
\tag{15}
$$





The vertical integration of the weak formulation, Eq. 6, requires the integration of the viscosity multiplied by a function of

$z$ only, $f(z)$, since the basis functions $\phi(x,y)$ employed in the approximation of the ice velocities $v_x$ and $v_y$ (Eq. 9) are depth-independent. The expression of $f(z)$ is a function of the polynomial function $1 - \zeta^{n+1}$ and varies according to the definitions of matrices $\mathbf{K}_{\mathrm{ij}}$ into the element matrix system (Eq. 12). The four expressions of $f(z)$ that appear in the element stiffness matrix are shown in Appendix C. Thus, we can write a generic expression for the vertical integration as follows:

$$\int_b^s \mu(x,y,z)\, f(z)\, dz. \tag{16}$$

In the following steps we drop the $x$ and $y$ dependency in the viscosity notation for simplicity. To achieve a vertically integrated formulation, let's assume that Eq. 16 can be rewritten by defining an *ad hoc* 'vertical averaged' viscosity $\bar{\mu} = \bar{\mu}(x,y)$ as follows:

$$\int_b^s \mu(z)f(z)dz = \bar{\mu}\int_b^s f(z)dz = \bar{\mu}F, \tag{17}$$

where $F$ is the primitive function of $f(z)$ evaluated between $b$ and $s$ (i.e., $F = [F(z)]_b^s$).

If Eq. 16 was evaluated by employing a Gaussian quadrature (between -1 and 1), the integral would be approximated by:

$$\int_b^s \mu(z)f(z)dz \approx \frac{H}{2}\sum \omega_i \mu(z_i)f(z_i), \tag{18}$$

with $\omega_i$ being the appropriate weights and $z_i$ the vertical coordinate evaluated at:

$$z_i = \frac{H}{2}\xi_i + \frac{s+b}{2},\ \xi_i \in [-1,1]. \tag{19}$$

By combining Eq. 17 and Eq. 18, we identify the *ad hoc* averaged viscosity as follows:

$$\bar{\mu} \approx \frac{H}{2F}\sum \omega_i \mu(z_i)f(z_i). \tag{20}$$

The accuracy of the assumption employed in Eq. 17 depends on the order of the numerical integration of Eq. 20. We test different number of integration points in Sect. A. Based on these numerical tests, we employ integration order equal to 5 to perform the ISMIP-HOM and MISMIP3D experiments. We note that the *ad hoc* viscosity $\bar{\mu}(x,y)$ varies within the triangular elements. In ISSM, we compute the value of $\bar{\mu}(x,y)$ through Eq. 20 at each quadrature point used during the two-dimensional

numerical integration of the triangular-element matrices $\mathbf{K}_{\mathrm{ij}}$ of Eq. 12. We also note that all the components of the effective strain rate tensor $\dot{\varepsilon}_e$ (Eq. 15) are evaluated using the horizontal velocities defined by Eq. 7.

## 3   Numerical experiments

In all numerical experiments performed here, a Picard iteration scheme is used to solve the nonlinear stress balance equations. We employ the Bi-Conjugate Gradient and Block-Jacobi precondition to solve the linear system at each Picard iteration.





**Table 1.** Constants used in the ISMIP-HOM experiment.

| symbol | description | value |
|:---:|:---:|:---:|
| $\rho$ | ice density | $910 \ \mathrm{kg \ m^{-3}}$ |
| $g$ | gravitational acceleration | $9.81 \ \mathrm{m \ yr^{-2}}$ |
| $A$ | ice rate factor | $10^{-16} \ \mathrm{Pa^{-3} yr^{-1}}$ |
| $n$ | Glen's law exponent | 3 |

## 3.1 ISMIP-HOM setup

To assess the performance of MOLHO, we first compare the velocities from HO and MOLHO based on the ISMIP-HOM benchmark (Pattyn et al., 2008). This benchmark aims to test the response of ice flow models in diagnostic runs using different scales of glacier geometries and flow regimes. Here, we perform experiments A, C, and E, for which only the ice velocities are computed (i.e., transient simulations are not considered). Table 1 shows the parameters employed in these experiments. We describe here the main characteristics of the experiments, and the details are found in Pattyn et al. (2008).

Experiment A consists of a flow over a sinusoidal bumpy bed with different *wavelengths*. The wavelengths (domain size) vary from 5 to 160 km. The amplitude of the bumps is 500 m. The bed is inclined in the $x$-direction. Basal velocities are set to zero and, therefore, this experiment is suitable to assess the performance of models in simulating internal deformation only (vertical shear). Periodic boundary conditions are imposed over the lateral boundaries of the model domain.

Experiment C is similar to experiment A. The bedrock is parallel to ice surface and has no bump. The ice base is allowed to slide over the bed following a Weertman-type sliding law:

$$\boldsymbol{\tau}^b = -C \parallel \boldsymbol{v}^b \parallel^{m-1} \boldsymbol{v}^b, \tag{21}$$

where $\boldsymbol{\tau}^b$ is the basal stress, $C$ is the friction coefficient, $\boldsymbol{v}^b$ is the ice base velocity, and $m$ is the sliding law exponent. In experiment C, $m = 1$ and the friction coefficient $C$ is defined by a sinusoidal function with different wavelengths that also vary from 5 to 160 km. This experiment assesses the performance of the model in stream-type flows.

The domain of experiment E is based on the 5-km center line of a real alpine glacier (Haut Glacier d'Arolla, Switzerland). The ice flows only in the $x$-direction and the width of the glacier ($y$-direction) is set to be constant and equal to 1 m over the entire glacier extent. The ice flow rate is uniform over the entire domain (see Table 1). Two basal boundary conditions are employed in this experiment: *frozen bed* and *slip bed*. In the former, basal velocities are set to zero and in the latter, a small region of perfect slip (i.e., $C = 0$) is imposed between $x$=2.2 km and $x$=2.5 km.

We employ the same horizontal mesh resolution in both HO and MOLHO models in all the experiments. To generate the three-dimensional mesh for HO, we extrude the horizontal mesh using 20 vertical layers. Due to convergence issues in the nonlinear solver, we use only 8 vertical layers to perform experiment E with HO.



## 3.2 MISMIP3D setup

The MISMIP3D benchmark (Pattyn et al., 2013) allows us to assess the convergence of the grounding line position at steady state, and the reversibility of the grounding line in the MOLHO flow model. In this experiment, an analytical solution for the steady state grounding line only exists for the Shelfy-Stream Approximation that does not account for vertical shear (Schoof, 2007a, b). Thus, for comparison purposes, we also run this benchmark with the three-dimensional HO model.

The setup consists of a marine ice sheet flowing along the $x$-axis over a uniformly sloping bedrock whose elevation, $r(x,y)$

(in m, negative if below sea level), is described by:

$$r(x,y) = 100 - x, \tag{22}$$

with $x$ in km. The ice sheet covers an area of $800 \times 50$ km$^2$. A no-flow condition ($v_x = 0$) is imposed at the ice divide ($x = 0$). A Neumann boundary condition based on ocean water pressure is applied at the calving front ($x = 800$ km). We impose a free-slip condition ($v_y = 0$) at the lateral boundaries of the domain ($y = 0$ and $y = 50$ km).

The basal sliding is described by a Weertman friction law as given by Eq. 21. The values of $C$ and $m$ as well as other parameters used in the experiments are shown in Table 3.

The MISMIP3D experiment is divided in three phases: (i) steady state (Stnd), (ii) basal friction perturbation (P75S), and (iii) basal friction restoration (P75R).

In phase Stnd, the model starts from a 100-m thick ice shelf and runs forward in time until a steady state is reached. Here,

the steady state is reached after 30,000 years of simulation.

The phase P75S starts from the end of phase Stnd. The friction coefficient $C$ is replaced by $C^*$ at the beginning of phase P75S. The new coefficient $C^*$ is defined as:

$$C^* = C \left[ 1 - 0.75 \times \exp \left( -\frac{(x - x_g)^2}{2x_c^2} - \frac{(y - y_g)^2}{2y_c^2} \right) \right]. \tag{23}$$

where $x_g$ is the the steady state grounding line position at $y = 0$, and $y_g = 0$. The spatial extent of the friction change along the

$x$ and $y$ directions is given by $x_c = 150$ km and $y_c = 10$ km, respectively. The model runs for 100 years with $C^*$. This friction perturbation forces the grounding line to migrate asymmetrically: it advances along $y = 0$ and retreats along $y = 50$ km.

At the end of phase P75S, the friction coefficient is restored to its initial value, $C$. Then, the model runs for 30,000 years to reach steady state again. This is the phase P75R. The grounding line reversibility is achieved if the positions of the grounding line at end of phases Stnd and P75R are sufficiently close to each other.

Previous intercomparison model results showed a strong mesh-resolution dependency to achieve grounding line reversibility (Pattyn et al., 2013; Feldmann et al., 2014). This dependency decreases if a sub-element friction parameterization is employed (Cornford et al., 2013; Feldmann et al., 2014; Seroussi et al., 2014). Here, we use sub-element parameterization type I (SEP1) as presented in Seroussi et al. (2014). To assess the robustness of the implementation of the MOLHO flow model, we run the three phases with different mesh resolutions (Table 2). The meshes are generated using Delaunay triangulation (Hecht,

2006), and a spatially uniform resolution is applied to most of them. To save computational resources, we generate the 250-m





**Table 2.** Mesh resolutions and associated number of elements used in the MISMIP3D experiment.

| Resolution | Number of elements | |
|---|---|---|
| | MOLHO | HO |
| 5 km | 3,278 | 32,780 |
| 2 km | 20,718 | 207,180 |
| 1 km | 84,300 | 843,000 |
| 500 m | 335,500 | 615,050 |
| *250 m | 656,621 | 2,079,660 |

*Used in the phase Stnd only.

**Table 3.** Constants used in the MISMIP3D experiment.

| symbol | description | value |
|---|---|---|
| $\rho$ | ice density | $900\,\text{kg m}^{-3}$ |
| $\rho_w$ | water density | $1\,000\,\text{kg m}^{-3}$ |
| $g$ | gravitational acceleration | $9.8\,\text{m yr}^{-2}$ |
| $\dot{m}_s$ | surface mass balance | $0.5\,\text{m yr}^{-1}$ |
| $C$ | friction coefficient | $10^7\,\text{Pa m}^{-1/3}\text{s}^{1/3}$ |
| $A$ | ice rate factor | $10^{-25}\,\text{Pa}^{-3}\text{s}^{-1}$ |
| $m$ | friction exponent | $1/3$ |
| $n$ | Glen's law exponent | $3$ |

resolution mesh by refining the 500-m mesh between $x = 400$ and $x = 650$ km. The HO model mesh is generated by extruding the two-dimensional mesh employed in the MOLHO model, except for the two finest resolutions (500 and 250 m) for which we coarsen all triangular elements located at $x < 500$ km and at $x > 620$ km before the mesh extrusion. We employ ten equally-spaced vertical layers in all HO model meshes. The number of elements for each model and mesh resolutions is shown

in Table 2. We note that the finest mesh (250 m) is used to analyse the convergence of the steady state grounding line (phase Stnd) only, since grounding line reversibility is already achieved with the coarser meshes (see Sect. 4.2).

## 4  Results

### 4.1  ISMIP-HOM

Figure 1 shows the Euclidean norm of the horizontal velocities $(v_x, v_y)$ at the ice surface and at $y = 0.25L$ for several wave-

lengths $(L)$ of bedrock bumps for experiment A. MOLHO and HO produce virtually the same surface speeds for wavelengths





equal or higher than 40 km. The difference between the speeds is about 4% for $L = 40$ km and about 2% for $L = 160$ km. For shorter bump lengths ($\leq 20$ km), MOLHO overestimates the ice speeds in comparison to HO. The differences are up to 60% for wavelength equal to 5 km. The differences are $\leq 11\%$ for $L = 20$ km.

In the case where the ice base slides over the bed (experiment C), MOLHO and HO yield similar surface velocities for all

wavelengths (Fig. 2). The differences between both models vary from 0.05% for $L = 5$ km to 1.2% for a $L = 160$ km. For comparison purposes, we also run experiment C with two other vertically integrated models that are also commonly employed for ice streams: a version of the L1L2 model (Schoof and Hindmarsh, 2010; Perego et al., 2012) and the Shelfy-Stream Approximation (SSA, Morland, 1987; MacAyeal, 1989). The L1L2 considers vertical shear in the ice viscosity computation, while the SSA model does not account for vertical shear stresses. For larger wavelengths of friction coefficient ($\geq 40$ km),

all four models yield similar velocities, as expected (Pattyn et al., 2008). The differences in speed between L1L2 and SSA in comparison to HO are up to 6% for $L = 40$ km and $<5\%$ for $L = 160$ km. For shorter wavelengths ($< 40$ km), vertical shear starts playing a role on the ice flow, which may compromise the results obtained with L1L2 and SSA models in comparison to HO. The maximum speed differences vary from 3 to 5% for $L = 5$ km and $L = 20$ km, respectively, for both L1L2 and SSA models. On the other hand, MOLHO 'approximates' the HO results even for the shortest $L$.

In the real glacier setup (experiment E) and considering a frozen bed, MOLHO overestimates the surface velocities up to 40%[1] over about the first 60% of the glacier flow line (Fig. 3). This region presents steeper bedrock than the region close to the glacier snout (see the glacier profile in Pattyn et al., 2008). Considering the whole domain, the RMSE (Root-Mean-Square Error) is 5.5 m yr$^{-1}$. Imposing a small region of perfect sliding in the model domain induces the glacier to speed-up. The difference in speeds achieved with MOLHO and HO is $< 40\%$[2]. The differences are higher over the first 3 km of the glacier,

as already noted in the frozen bed setup. The RMSE is 7.6 m yr$^{-1}$.

### 4.2 MISMIP3D

The steady state positions of the grounding line (GL) at $y = 0$ for different mesh resolutions are shown in Fig. 4. We also present the grounding line positions obtained with the HO model. To estimate the convergence error in GL positions as we increase the mesh resolution, we compute the relative error of the steady state GL position $x_g^h$ obtained with a mesh resolution

$h$, defined as follows:

$$\varepsilon_{x_g}^h = \left| \frac{x_g^{2h} - x_g^h}{x_g^{2h}} \right|, \qquad (24)$$

where $x_g^{2h}$ is the position of the grounding line obtained with the closest coarser mesh resolution. The estimated convergence errors for both MOLHO and HO models are also shown in Fig. 4.

Using a coarser mesh resolution (5 km), the grounding lines obtained with both HO and MOLHO extend a few kilometers

beyond 600 km, reaching the steady state GL location predicted by the Boundary Layer analysis and achieved by numerical simulations using SSA flow models (Schoof, 2007b; Cornford et al., 2013; Feldmann et al., 2014; Seroussi et al., 2014).

---

[1]Excluding the first 15% of the glacier domain since the velocities are relatively small there.

[2]See footnote 1.

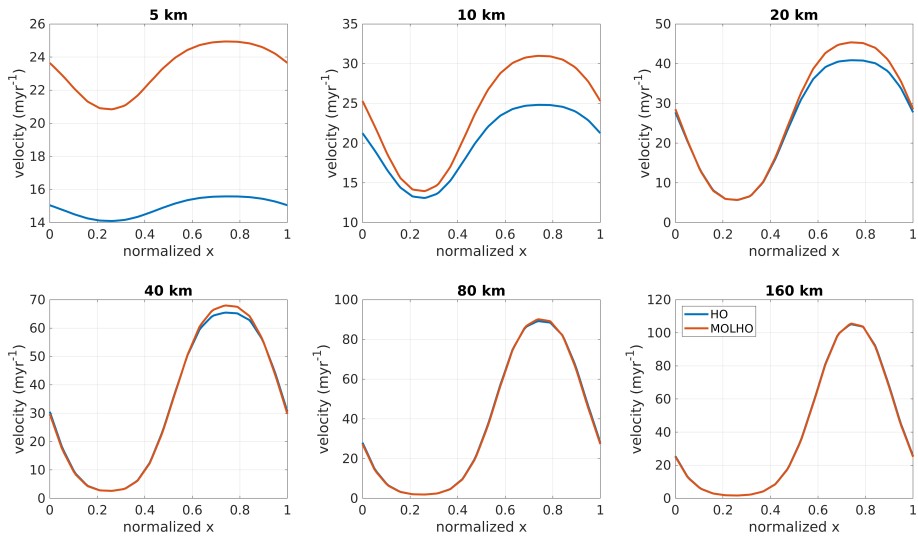

**Figure 1.** ISMIP-HOM experiment A: surface velocity at $y = 0.25L$ for different wavelengths, $L$, obtained with the tree-dimensional Higher-Order model (HO) and with the two-dimensional MOno-Layer Higher-Order model (MOLHO). The surface velocity is computed by taking the Euclidean norm of the horizontal velocities at the ice surface (i.e., $\sqrt{v_x^2 + v_y^2}$). We employ 20 vertical layers for the HO model.

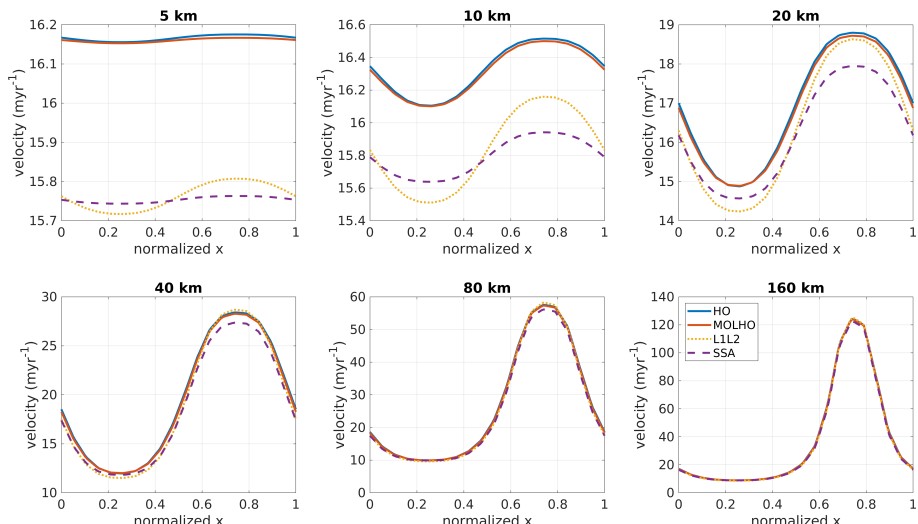

**Figure 2.** ISMIP-HOM experiment C: surface velocity at $y = 0.25L$ for different wavelengths, $L$, obtained with the tree-dimensional Higher-Order model (HO) and with the two-dimensional MOno-Layer Higher-Order model (MOLHO). Results obtained with L1L2 and Shelfy-Stream Approximation (SSA) models are also shown. The surface velocity is computed by taking the Euclidean norm of the horizontal velocities at the ice surface (i.e., $\sqrt{v_x^2 + v_y^2}$). We employ 20 vertical layers for the HO model.

However, as obtained by other ice sheet models employing Full-Stokes, Hybrid, or Higher-Order flow equations, the grounding





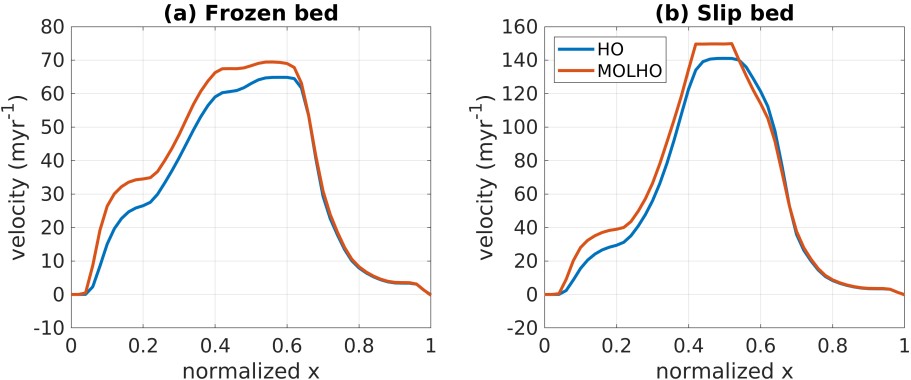

**Figure 3.** ISMIP-HOM experiment E: surface velocity in the ice flow direction obtained with the tree-dimensional Higher-Order model (HO) and with the two-dimensional Mono-Layer Higher-Order model (MOLHO). Two basal conditions are presented: frozen bed (a) and slip bed (b). We employ 8 vertical layers for the HO model.

line at steady state tends to be located between 520 and 580 km (e.g., Pattyn et al., 2013; Cornford et al., 2013; Gagliardini et al., 2016). As seen in Fig. 4, the grounding lines obtained with MOLHO and HO approach to 560 km as the mesh gets finer. 265 With the finest mesh used here (250 m), the steady state GL positions computed with MOLHO and HO models are 562.100 km and 559.916 km, respectively, a difference of less than 0.5% between them.

As also shown in Fig. 4, both HO and MOLHO flow models show similar convergence of the relative error in GL position. The relative errors decrease with mesh resolution, reaching about 1% with a mesh resolution of 250 m.

At the end of the restoration phase (P75R), the GL position is similar to the position obtained in the steady state phase (Stnd, 270 Fig. 5), which indicates that grounding line reversibility is fully achieved with MOLHO. Overall, the simulated displacement of the grounding line in the perturbation phase (P75S) using MOLHO is comparable to the ones simulated with the HO model, as seen in Fig. 5. At $y = 0$, the GL advances about 11-13 km, while at $y = 50$ km the GL retreats 4-6 km (Table 4). As shown in Table 4, the displacements of the grounding line in the P75S experiment using MOLHO are in good agreement with the ones obtained with HO.

The transient response of the grounding line at $y = 0$ km and $y = 50$ km during the P75S experiment and the first 100 years of the restoration phase (P75R) is shown in Fig. 6. As shown in the figure, the perturbation and restoration responses of the grounding obtained with MOLHO and HO are similar.

To analyze the differences in ice velocity in MOLHO and HO, we perform diagnostic runs employing the same MISMIP3D-type ice sheet profile. For this run, we use the ice geometry (surface, base, thickness, and grounding line) obtained with HO in 280 the phase Stnd and with a mesh resolution of 1 km. In Figure 7 we show the surface and basal velocities around the grounding line, where there are more pronounced differences between the model results. We also show the depth-averaged velocity (Eq. 8 for MOLHO model), which controls the advection of ice downstream. Interestingly, for this ice sheet profile, we find that MOLHO produces slightly lower velocities compared to HO (Fig. 7), which may explain why the steady state grounding lines





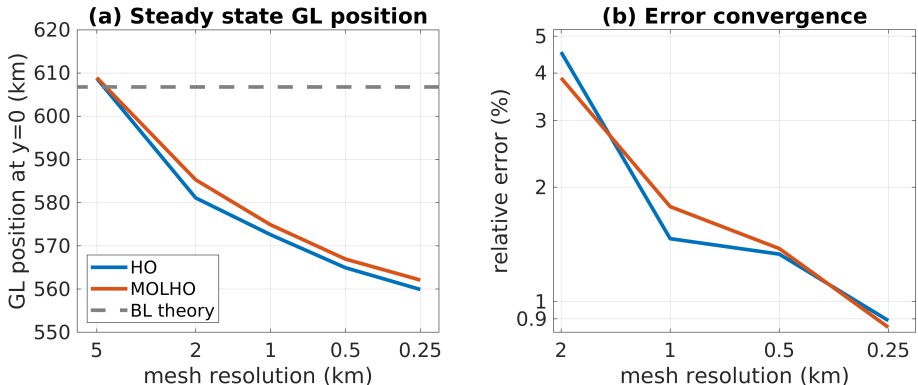

**Figure 4.** MISMIP3D experiment: steady state positions (a) and estimated error convergence (b) of the grounding line (GL) at $y = 0$ obtained with different mesh resolutions using HO and MOLHO. The error convergence is estimated using Eq. 24. For comparison, the steady state grounding line position predicted by the Boundary Layer (BL) theory that is based on SSA model is also shown.

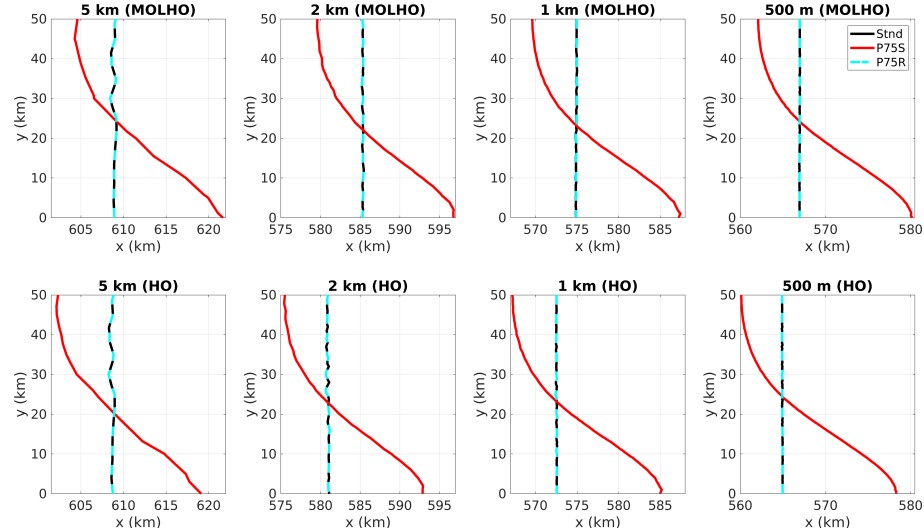

**Figure 5.** Grounding line positions of the MISMIP3D experiments obtained with different mesh resolutions. Black lines are the steady state grounding line positions (Stnd). Red lines are the positions obtained after 100 years of basal friction perturbation (P75S). Dashed blue lines are the new steady state grounding line positions obtained after resetting the friction coefficient to its original value (P75R).

are slightly advanced in comparison to the HO model simulations (Fig. 5). The differences over the region shown in Fig. 7 are
up to 4.5 km yr$^{-1}$, which is less than 1.5%.

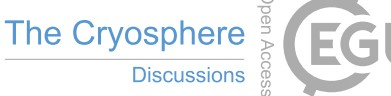

**Table 4.** Displacements of the grounding line ($\Delta$GL) at the end of the perturbation phase (P75S) obtained with MOLHO and HO models.

| Resolution | MOLHO | HO |
|---|---|---|
| $\Delta$ GL ($y = 0$ km) | | |
| 5 km | 12.7 km | 10.4 km |
| 2 km | 11.4 km | 11.8 km |
| 1 km | 12.3 km | 12.4 km |
| 500 m | 13.2 km | 13.4 km |
| $\Delta$ GL ($y = 50$ km) | | |
| 5 km | -4.5 km | -6.5 km |
| 2 km | -5.7 km | -5.4 km |
| 1 km | -5.3 km | -5.3 km |
| 500 m | -4.9 km | -4.8 km |

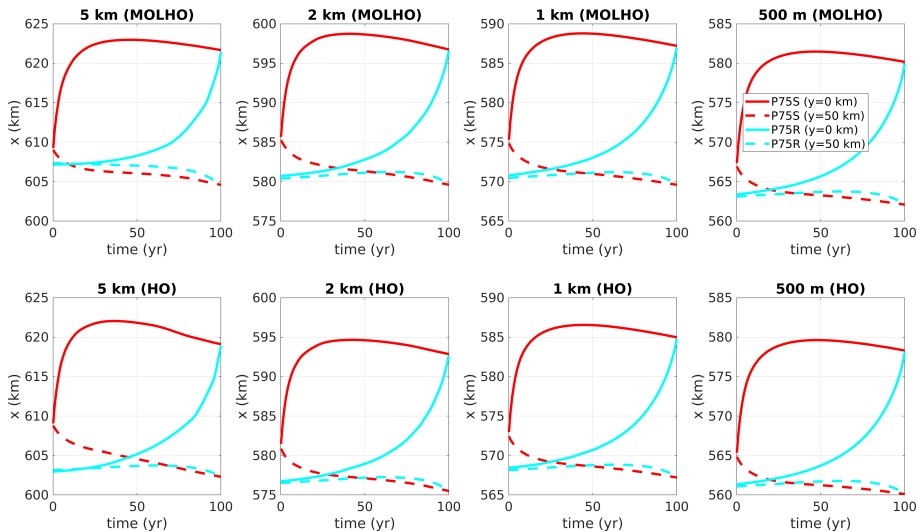

**Figure 6.** Time-dependent grounding line positions of the MISMIP3D experiments obtained with different mesh resolutions. Solid lines are the grounding line positions at $y = 0$. Dashed lines are the grounding line positions at $y = 50$ km. Red lines are the positions obtained during the basal friction perturbation experiment (P75S). Blue lines are the positions obtained during the first 100 years after restoring the friction coefficient of its original value (P75R). Note that the $x$ axes are reversed for the P75R experiment results.



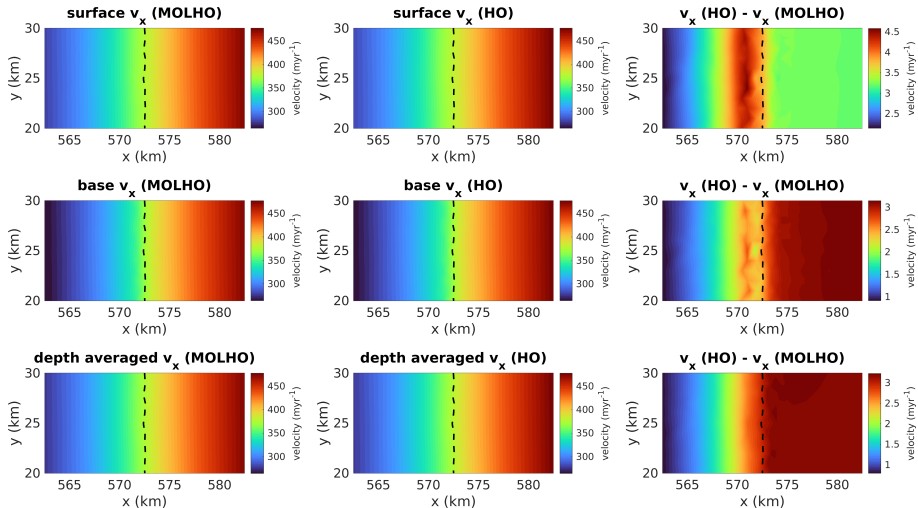

**Figure 7.** Comparison of ice velocities in the $x$-direction obtained with MOLHO and HO using the same glacier geometry (thickness and grounding line). The figure shows the velocities at the surface and base of the glacier as well as the depth-averaged velocities. The velocities are computed using the steady state geometry obtained with the phase Stnd with HO model and horizontal mesh resolution of 1 km. Dashed lines are the grounding line positions.

### 4.3 Time performance

To assess and illustrate the computational cost of MOLHO in time-dependent ice sheet simulations, we run the first 500 years of the spin-up phase of the MISMIP3D setup (Stnd). This phase is long enough to simulate a large grounding line migration (advance, in this case), therefore requiring several iterations of the stress balance solver.

We use four different mesh resolutions (5, 2, 1, and 0.5 km) and three different ice flow approximations: MOLHO, HO, and SSA. For this time performance comparison, we use the same MOLHO mesh for both HO and SSA. For HO, we extrude the horizontal mesh using ten vertical layers. The number of elements of each MOLHO mesh is shown in Table 2, while Table 5 shows the total number of degrees of freedom for each mesh and flow model. The experiments are performed in parallel using 28 computational cores (CPUs) in a 2× Intel Xeon E5-2680v4 2.8 GHz with 128 GB of memory. The time step is 0.125 year, which fulfills the CFL condition for all meshes.

As shown in Table 5, the computation time using MOLHO is, overall, one order of magnitude lower in comparison to the time spent in HO. The ratio between the simulation times obtained with MOLHO and HO varies with mesh resolution. For coarser meshes, MOLHO is 10× faster than HO. For the finer mesh resolutions, MOLHO is almost 20× faster than HO. The number of degrees of freedom for the HO model is 5.5× higher than MOLHO, for all mesh resolutions employed here.



**Table 5.** Computational time to run the first 500 years of the MISMIP3D spin-up phase (Stnd). For comparison purposes, we show the simulation times obtained by three different ice flow models: MOLHO, HO, and SSA. SSA and HO employs the same mesh as MOLHO. For HO, we use ten vertical layers. The number of elements for each mesh resolution employed in MOLHO/SSA is shown in Table 2.

| Resolution | SSA | MOLHO | HO |
|---|---|---|---|
| | Simulation time (s) | | |
| 5 km | 394 | 570 | 6,047 |
| 2 km | 1,324 | 2,498 | 47,580 |
| 1 km | 4,752 | 16,223 | 232,904 |
| 500 m | 58,341 | 102,948 | *2,000,000 |
| | Degrees of freedom | | |
| 5 km | 3,620 | 7,240 | 39,820 |
| 2 km | 21,570 | 43,140 | 237,270 |
| 1 km | 86,002 | 172,004 | 946,022 |
| 500 m | 338,902 | 677,804 | 3,727,922 |

*Estimated.

In comparison to SSA, MOLHO is between 1.5 to 3.5× slower, depending on the mesh resolution. This is probably due to different rates of convergence of the linear solver during the Picard iterations. MOLHO has, by construction, 2× more degrees of freedom than SSA for any given mesh, which demands *theoretically* 4× more[3] computational time.

## 5 Discussion

By running the three phases of the MISMIP3D setup, we obtain grounding line reversibility for all mesh resolutions employed here. The evolution of the grounding line during the friction perturbation and restoration phases (P75S and P75R, respectively) are consistent with previous intercomparison results (Pattyn et al., 2013). These results confirm the robustness of our implementation and the ability of MOLHO to simulate grounding line dynamics in marine ice sheets at a similar accuracy as HO.

Using the MOLHO flow model, the convergence of the steady state grounding line positions with mesh resolutions follows the ones achieved with the three-dimensional Higher-Order model. This suggests that the ice velocity computation and the friction parameterizations employed in MOLHO are consistent with the HO model implemented in ISSM. Yet, there is still room to improve the convergence rate of both MOLHO and HO models with mesh resolution.

Overall, MOLHO's results (ice velocities and grounding line positions) are in close agreement with the results obtained with a full three-dimensional model. This is especially true for the marine ice sheet with fast sliding (e.g., MISMIP3D). Similar

---

[3]Considering an iterative linear solver with time complexity of $\mathcal{O}\left(n^2\right)$, where $n$ is the number of degrees of freedom.





results are also observed in the ISMIP-HOM (Pattyn et al., 2008) experiment C where basal sliding is present (see Fig. 2). In experiment A, MOLHO overestimates speed relative to the HO for deformation dominated cases (shorter wavelengths of bedrock elevation). Our computation of the vertically averaged ice viscosity $\bar{\mu}$ might be the reason behind some of the velocity overestimation, since the longitudinal stresses present in the upper layers of the ice are being 'watered down' by the averaging with the softer ice at the bed. This suggests that the choice of the vertical quadrature scheme (e.g., order and/or vertical

distribution of Gaussian points) plays an important role in the results of the MOLHO formulation (Brinkerhoff and Johnson, 2015). Also, the determinant of the Jacobian in HO model is not constant within the prismatic elements (due to the mapping between the 'real' and reference elements), which might be another source to explain these differences.

  In terms of time performance, the computational cost of MOLHO is relatively higher than the SSA model. This is expected since MOLHO has $2\times$ more degrees of freedom than SSA, and also includes a vertical integration of the ice viscosity at every

Gauss point when we compute the element stiffness matrix. In comparison to HO with ten vertical layers employed in the time performance analysis, the computational cost is at least $10\times$ lower. While it is possible to decrease the simulation time of HO by decreasing the number of vertical layers (at the risk of reducing numerical accuracy) and by improving the scalability (i.e., increasing the number of CPUs and/or computational nodes), there will always be bottlenecks related to three-dimensional meshes that will increase computational cost in comparison to two-dimensional meshes.

In MOLHO, the vertical variation of the horizontal velocities is prescribed by a SIA-derived polynomial function of order equal to $n+1$. In the experiments performed here, we assumed isothermal ice flow. However, the vertical integration of the ice viscosity can accommodate vertical variation in ice temperature by changing the polynomial function accordingly, e.g., solving for the analytical SIA solution and using a scaled version of that as the vertical basis function.

## 6 Conclusions

In this work, we presented the formulation of a vertically integrated, MOno-Layer Higher-Order model, and compared its performance with the three-dimensional Blatter-Pattyn model using two benchmarks: ISMIP-HOM and MISMIP3D. In the experiments with no basal sliding of the ISMIP-HOM setup (Exp. A), MOLHO produces surface velocities close to the ones obtained by HO model for wavelengths of bedrock elevation equal or higher than 40 km. For shorter wavelengths (<40 km) where the approximation in vertical shear may break down, MOLHO tends to overestimates surface velocities. When the ice

base is allowed to slide (Exp. C), MOLHO and HO produce similar surface velocities for all lengths of bed undulation. These results suggest that MOLHO is more suitable to simulate larger ice masses such as marine ice sheets, than small ice caps. Yet, for the Haut Glacier d'Arolla experiment (Exp. E), MOLHO overestimates upstream velocities in up to 40%, but produces similar speed over the downstream part of the glacier profile. In the MISMIP3D experiments, the steady state grounding line positions obtained with MOLHO are very close to the positions obtained with HO model. The grounding line migrations have

a similar temporal behavior. This suggests that the grounding line dynamics achieved with MOLHO are consistent with the original HO model. By carrying out a time performance analysis in time-dependent runs, we found that the computation cost





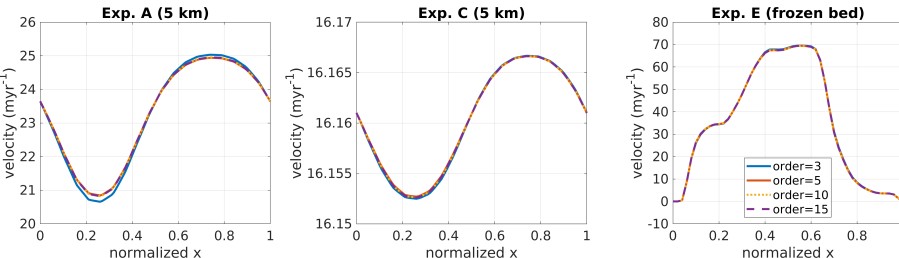

**Figure A1.** ISMIP-HOM experiments: surface velocities for experiments A, C and E obtained with the MOLHO model using different integration orders for the ice viscosity computation. Only the short wavelength (5 km) case is shown for experiments A and C. For experiment E, only the frozen bed case is shown.

of MOLHO is at least $10\times$ lower in comparison to HO model. MOLHO is therefore an excellent alternative to HO for long simulations, such as paleo ice sheet modeling.

*Code availability.* The MOno-Layer Higher-Order (MOLHO) model evaluated here is currently implemented in ISSM. The code can be
downloaded, compiled and executed following the instructions available on the ISSM website: https://issm.jpl.nasa.gov/download (last access: 26 August 2021). The public SVN repository for the ISSM code can also be found directly at https://issm.ess.uci.edu/svn/issm/issm/trunk (Larour et al., 2020). The version of the code for this study, corresponding to ISSM release 4.18, is SVN version tag number 26413. The documentation of the code version used here is available at https://issm.jpl.nasa.gov/documentation/ (last access: 26 August 2021).

## Appendix A: Integration order of the ice viscosity

The accuracy of the semi-analytic integration of the vertical axis in MOLHO formulation depends on the integration order employed in the ice viscosity computation (Eq. 20). Figure A1 shows the surface speeds for the ISMIP-HOM experiments obtained with four orders tested: 3, 5, 10, and 15. We only present the cases for the shortest wavelength (5 km) for experiments A and C, and the frozen bed case, for experiment E.

For experiments A and C, the differences between the velocities obtained with an integration order equal to 3 and equal to
15 are smaller than 1%. The difference reduces to 0.1% when we compare orders 5 and 15. Integrating orders 10 and 15 yield similar surface velocities: the differences are smaller than 0.0005%.

For experiment E, the speed differences are up to 3% comparing integration orders 3 and 15; between 5 and 15, the differences reduce to 0.5%. Comparing orders 10 and 15, the speed differences are smaller than 0.005%.



## Appendix B:  Useful identities

Using the definition of $\zeta$ given by Eq. 3, we use the following identities:

$$\frac{\partial}{\partial z}\left(1-\zeta^{n+1}\right) = \frac{n+1}{H}\zeta^n, \tag{B1}$$

$$\int_b^s \zeta^{n+1}dz = \frac{H}{(n+2)}, \tag{B2}$$

$$\int_b^s \left(1-\zeta^{n+1}\right)dz = H\frac{(n+1)}{(n+2)}, \tag{B3}$$

$$\int_b^s \left(1-\zeta^{n+1}\right)^2 dz = 2H\frac{(n+1)^2}{(2n+3)(n+2)}, \tag{B4}$$

$$\int_b^s \left(\frac{\partial}{\partial z}\left(1-\zeta^{n+1}\right)\right)^2 dz = \frac{(n+1)^2}{H(2n+1)}, \tag{B5}$$

$$\int_b^s \zeta\left(1-\zeta^{n+1}\right)dz = \frac{H}{2}\frac{(n+1)}{(n+3)}, \tag{B6}$$

$$\int_b^0 \zeta^{n+1}dz = \frac{H}{n+2}\left[1-\left(\frac{s}{H}\right)^{n+2}\right], \tag{B7}$$

$$\int_b^0 \zeta^{n+2}dz = \frac{H}{(n+3)}\left[1-\left(\frac{s}{H}\right)^{n+3}\right], \tag{B8}$$

$$\int_b^0 z\left(1-\zeta^{n+1}\right)dz = -\frac{b^2}{2} - \frac{sH}{n+2}\left[1-\left(\frac{s}{H}\right)^{n+2}\right] + \frac{H^2}{(n+3)}\left[1-\left(\frac{s}{H}\right)^{n+3}\right]. \tag{B9}$$





## Appendix C: Vertically integrated ice viscosity

The function $f(z)$ can have the following expressions (see Appendix D):

$$f(z) = \begin{cases} f_1(z) & = 1, \\ f_2(z) & = \left(1 - \zeta^{n+1}\right), \\ f_3(z) & = \left(1 - \zeta^{n+1}\right)^2, \\ f_4(z) & = \left(\frac{\partial}{\partial z}\left(1 - \zeta^{n+1}\right)\right)^2, \end{cases} \tag{C1}$$

where we keep $\zeta$ (Eq. 3) to simplify the notation. The vertical integration of the primitive of $f$, $F$, reads:

$$F = \begin{cases} F_1 & = H, \\ F_2 & = H\frac{(n+1)}{(n+2)}, \\ F_3 & = 2H\frac{(n+1)^2}{(2n+3)(n+2)}, \\ F_4 & = \frac{(n+1)^2}{H(2n+1)}. \end{cases} \tag{C2}$$

Therefore, we have four expressions for the vertically integrated *ad hoc* ice viscosity $\bar{\mu}$ that are used in different parts of the element stiffness matrix (see Appendix D):

$$\bar{\mu} = \begin{cases} \bar{\mu}_1 & = \frac{H}{2F_1}\sum \omega_i \mu(z_i) f_1(z_i), \\ \bar{\mu}_2 & = \frac{H}{2F_2}\sum \omega_i \mu(z_i) f_2(z_i), \\ \bar{\mu}_3 & = \frac{H}{2F_3}\sum \omega_i \mu(z_i) f_3(z_i), \\ \bar{\mu}_4 & = \frac{H}{2F_4}\sum \omega_i \mu(z_i) f_4(z_i). \end{cases} \tag{C3}$$

## Appendix D: Stiffness matrix

By inserting Eq. 9 and Eq. 11 into the weak formulation Eq. 6, and by employing the useful identities shown in Appendix B and the four expressions of the *ad hoc* ice viscosity defined in Appendix C, we integrate the terms of the element stiffness matrix along the vertical axis. In the following integrals, $\Omega_e$ is the two-dimensional element domain defined over the horizontal $xy$-plane (e.g., triangle, for Delaunay triangulation-based meshes).

$$\mathbf{K}_{11} = \left[ \int_{\Omega_e} 4H\bar{\mu}_1 \frac{\partial \phi_j}{\partial x}\frac{\partial \phi_i}{\partial x} + H\bar{\mu}_1 \frac{\partial \phi_j}{\partial y}\frac{\partial \phi_i}{\partial y} \, d\Omega_e \right]$$





$$\mathbf{K}_{12} = \left[ \int_{\Omega_e} \left( 4H\bar{\mu}_2 \frac{\partial\phi_j}{\partial x}\frac{\partial\phi_i}{\partial x} + H\bar{\mu}_2 \frac{\partial\phi_j}{\partial y}\frac{\partial\phi_i}{\partial y} \right) \frac{(n+1)}{(n+2)} \, d\Omega_e \right]$$

$$\mathbf{K}_{13} = \left[ \int_{\Omega_e} 2H\bar{\mu}_1 \frac{\partial\phi_j}{\partial y}\frac{\partial\phi_i}{\partial x} + H\bar{\mu}_1 \frac{\partial\phi_j}{\partial x}\frac{\partial\phi_i}{\partial y} \, d\Omega_e \right]$$

$$\mathbf{K}_{14} = \left[ \int_{\Omega_e} \left( 2H\bar{\mu}_2 \frac{\partial\phi_j}{\partial y}\frac{\partial\phi_i}{\partial x} + H\bar{\mu}_2 \frac{\partial\phi_j}{\partial x}\frac{\partial\phi_i}{\partial y} \right) \frac{(n+1)}{(n+2)} \, d\Omega_e \right]$$

$$\mathbf{K}_{21} = \left[ \int_{\Omega_e} \left( 4H\bar{\mu}_2 \frac{\partial\phi_j}{\partial x}\frac{\partial\phi_i}{\partial x} + H\bar{\mu}_2 \frac{\partial\phi_j}{\partial y}\frac{\partial\phi_i}{\partial y} \right) \frac{(n+1)}{(n+2)} \, d\Omega_e \right]$$

$$\mathbf{K}_{22} = \left[ \int_{\Omega_e} \left( 4H\bar{\mu}_3 \frac{\partial\phi_j}{\partial x}\frac{\partial\phi_i}{\partial x} + H\bar{\mu}_3 \frac{\partial\phi_j}{\partial y}\frac{\partial\phi_i}{\partial y} \right) \frac{2(n+1)^2}{(2n+3)(n+2)} + \bar{\mu}_4 \phi_j \phi_i \frac{(n+1)^2}{H(2n+1)} \, d\Omega_e \right]$$

$$\mathbf{K}_{23} = \left[ \int_{\Omega_e} \left( 2H\bar{\mu}_2 \frac{\partial\phi_j}{\partial y}\frac{\partial\phi_i}{\partial x} + H\bar{\mu}_2 \frac{\partial\phi_j}{\partial x}\frac{\partial\phi_i}{\partial y} \right) \frac{(n+1)}{(n+2)} \, d\Omega_e \right]$$

$$\mathbf{K}_{24} = \left[ \int_{\Omega_e} \left( 2H\bar{\mu}_3 \frac{\partial\phi_j}{\partial y}\frac{\partial\phi_i}{\partial x} + H\bar{\mu}_3 \frac{\partial\phi_j}{\partial x}\frac{\partial\phi_i}{\partial y} \right) \frac{2(n+1)^2}{(2n+3)(n+2)} \, d\Omega_e \right]$$

$$\mathbf{K}_{31} = \left[ \int_{\Omega_e} 2H\bar{\mu}_1 \frac{\partial\phi_j}{\partial x}\frac{\partial\phi_i}{\partial y} + H\bar{\mu}_1 \frac{\partial\phi_j}{\partial y}\frac{\partial\phi_i}{\partial x} \, d\Omega_e \right]$$

$$\mathbf{K}_{32} = \left[ \int_{\Omega_e} \left( 2H\bar{\mu}_2 \frac{\partial\phi_j}{\partial x}\frac{\partial\phi_i}{\partial y} + H\bar{\mu}_2 \frac{\partial\phi_j}{\partial y}\frac{\partial\phi_i}{\partial x} \right) \frac{(n+1)}{(n+2)} \, d\Omega_e \right]$$

$$\mathbf{K}_{33} = \left[ \int_{\Omega_e} 4H\bar{\mu}_1 \frac{\partial\phi_j}{\partial y}\frac{\partial\phi_i}{\partial y} + H\bar{\mu}_1 \frac{\partial\phi_j}{\partial x}\frac{\partial\phi_i}{\partial x} \, d\Omega_e \right]$$





$$\mathbf{K}_{34} = \left[ \int\limits_{\Omega_e} \left( 4H\bar{\mu}_2 \frac{\partial \phi_j}{\partial y} \frac{\partial \phi_i}{\partial y} + H\bar{\mu}_2 \frac{\partial \phi_j}{\partial x} \frac{\partial \phi_i}{\partial x} \right) \frac{(n+1)}{(n+2)} \, d\Omega_e \right]$$

$$\mathbf{K}_{41} = \left[ \int\limits_{\Omega_e} \left( 2H\bar{\mu}_2 \frac{\partial \phi_j}{\partial x} \frac{\partial \phi_i}{\partial y} + H\bar{\mu}_2 \frac{\partial \phi_j}{\partial y} \frac{\partial \phi_i}{\partial x} \right) \frac{(n+1)}{(n+2)} \, d\Omega_e \right]$$

$$\mathbf{K}_{42} = \left[ \int\limits_{\Omega_e} \left( 2H\bar{\mu}_3 \frac{\partial \phi_j}{\partial x} \frac{\partial \phi_i}{\partial y} + H\bar{\mu}_3 \frac{\partial \phi_j}{\partial y} \frac{\partial \phi_i}{\partial x} \right) \frac{2(n+1)^2}{(2n+3)(n+2)} \, d\Omega_e \right]$$

$$\mathbf{K}_{43} = \left[ \int\limits_{\Omega_e} \left( 4H\bar{\mu}_2 \frac{\partial \phi_j}{\partial y} \frac{\partial \phi_i}{\partial y} + H\bar{\mu}_2 \frac{\partial \phi_j}{\partial x} \frac{\partial \phi_i}{\partial x} \right) \frac{(n+1)}{(n+2)} \, d\Omega_e \right]$$

$$\mathbf{K}_{44} = \left[ \int\limits_{\Omega_e} \left( 4H\bar{\mu}_3 \frac{\partial \phi_j}{\partial y} \frac{\partial \phi_i}{\partial y} + H\bar{\mu}_3 \frac{\partial \phi_j}{\partial x} \frac{\partial \phi_i}{\partial x} \right) \frac{2(n+1)^2}{(2n+3)(n+2)} + \bar{\mu}_4 \phi_j \phi_i \frac{(n+1)^2}{H(2n+1)} \, d\Omega_e \right]$$

## Appendix E: Stiffness matrix - basal friction

In ISSM, basal friction is written as a function of the basal velocities $(v_x^b, v_y^b)$. Therefore, the matrices $\mathbf{K}_{ii}^b$ $(i = 1, 3)$ below should be added to the respective stiffness matrices. The parameter $\alpha^2 = \alpha^2 \left( x, y, v_x^b, v_y^b \right)$ depends on the friction law employed.

$$\mathbf{K}_{11}^b = \left[ \int\limits_{\Omega_e} \alpha^2 \phi_j \phi_i \, d\Omega_e \right]$$

$$\mathbf{K}_{33}^b = \left[ \int\limits_{\Omega_e} \alpha^2 \phi_j \phi_i \, d\Omega_e \right]$$

## Appendix F: Loading vector - driving stress

The driving stress is integrated over the element domain, $\Omega_e$. The resulting loading vectors are:

$$\boldsymbol{F}_1 = \left[ -\int\limits_{\Omega_e} \rho g H \frac{\partial s}{\partial x} \phi_i \, d\Omega_e \right]$$





$$\boldsymbol{F}_2 \;=\; \left[\; -\int_{\Omega_e} \rho g H \frac{\partial s}{\partial x} \phi_i \frac{(n+1)}{(n+2)} \, d\Omega_e \;\right]$$

$$\boldsymbol{F}_3 \;=\; \left[\; -\int_{\Omega_e} \rho g H \frac{\partial s}{\partial y} \phi_i \, d\Omega_e \;\right]$$


$$\boldsymbol{F}_4 \;=\; \left[\; -\int_{\Omega_e} \rho g H \frac{\partial s}{\partial y} \phi_i \frac{(n+1)}{(n+2)} \, d\Omega_e \;\right]$$

**Appendix G: Loading vector - calving front**

If an element edge $\Gamma_e^w$ is at the calving front, the vectors $\mathbf{F}_i^w$ $(i=1,\ldots,4)$ below should be added to the element loading vector. In the following integrals, $n_x$ and $n_y$ are the components of the unit vector pointing outward edge $\Gamma_e^w$.

$$\boldsymbol{F}_1^w \;=\; \left[\; \int_{\Gamma_e^w} \left( \frac{1}{2}\rho g H^2 - \frac{1}{2}\rho_w g b^2 \right) n_x \phi_i \, d\Gamma_e^w \;\right]$$


$$\boldsymbol{F}_2^w \;=\; \left[\; \int_{\Gamma_e^w} \rho_w g \left\{ -\frac{b^2}{2} - \frac{sH}{n+2}\left[1-\left(\frac{s}{H}\right)^{n+2}\right] + \frac{H^2}{(n+3)}\left[1-\left(\frac{s}{H}\right)^{n+3}\right] \right\} n_x \phi_i + \rho g H \left[ \frac{H}{2}\frac{(n+1)}{(n+3)} \right] n_x \phi_i \, d\Gamma_e^w \;\right]$$

$$\mathbf{F}_3^w \;=\; \left[\; \int_{\Gamma_e^w} \left( \frac{1}{2}\rho g H^2 - \frac{1}{2}\rho_w g b^2 \right) n_y \phi_i \, d\Gamma_e^w \;\right]$$

$$\boldsymbol{F}_4^w \;=\; \left[\; \int_{\Gamma_e^w} \rho_w g \left\{ -\frac{b^2}{2} - \frac{sH}{n+2}\left[1-\left(\frac{s}{H}\right)^{n+2}\right] + \frac{H^2}{(n+3)}\left[1-\left(\frac{s}{H}\right)^{n+3}\right] \right\} n_y \phi_i + \rho g H \left[ \frac{H}{2}\frac{(n+1)}{(n+3)} \right] n_y \phi_i \, d\Gamma_e^w \;\right]$$

*Author contributions.* DB proposed the idea of a single layer of a higher order ice flow model. TDS and MM worked on the finite element formulation and implemented the MOno Layer Higher Order (MOLHO) model in ISSM. TDS designed the experimental setup and performed
the simulations. TDS and MM led the analysis of the results. TDS led the initial writing of the paper. All authors contributed to writing the final version of the paper.



*Competing interests.* The authors declare that they have no conflict of interest.

*Acknowledgements.* This work is from the PROPHET project, a component of the International Thwaites Glacier Collaboration (ITGC). Support from National Science Foundation (NSF: Grant 1739031)]. ITGC Contribution No. ITGC-054.



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
