# Peer review of "A new vertically integrated, MOno-Layer Higher-Order ice flow model (MOLHO)"

_The Cryosphere, 2021_

## Author Comment (AC1)

**Author's response for the manuscript "A new vertically integrated, MOno-Layer Higher-Order ice flow model (MOLHO)"**

Thiago Dias dos Santos, Mathieu Morlighem, and Douglas Brinkerhoff

On behalf of my co-authors, I would like to thank the Editor, Benjamin Smith, for handling the review process of our manuscript. We also thank Daniel Shapero (RC1) and the anonymous reviewer (RC2) for their positive and constructive comments. The comments helped improve the manuscript and clarify some important points.

Please, see below our responses (in **blue**) for each comment (in **black**).

**Response to Daniel Shapero (RC1)**

This paper further explores the design space of cost-efficient ice flow models that can capture both plug and shear flow. The more faithful but more expensive model has been known for a long time -- the first-order or Blatter-Pattyn equations, derived through a perturbative simplification of the full Stokes equations. Several authors have proposed further simplifications of the Blatter-Pattyn equations using a variety of techniques, for example the multi-layer shallow shelf approximation of Jouvet et al. (2015), which works by a finite difference semi-discretization in the vertical dimension. This paper proceeds instead by using a Galerkin-type semi-discretization in the vertical dimension. This idea has several precedents in the literature; the only citation that I'd add is Langdon and Raymond (1978), which is the earliest reference I can find of anyone trying this idea.

**We thank Daniel Shapero (RC1) for his constructive comments and suggestions which helped improve the clarity of the paper. As suggested by the reviewer, we added the reference "Langdon and Raymond (1978)".**

This paper does a good job illustrating the differences between the model they implement and closely related models that have appeared elsewhere in the literature, e.g. Brinkerhoff and Johnson (2015) and the work of myself and coauthors this year. Assuming that this model is part of ISSM or will be merged, this paper is also valuable as documentation of the underlying mathematics for future users. Having implemented a model in a similar spirit, I found the vertical averaging they used to be a convincing approach for keeping the computational cost down while preserving as much physical accuracy as possible. I also found their observation that the model has a bias for faster flow in ice sheet interiors to be especially important and motivating of future work that might correct this problem.

**Thank you for supporting the publication of this manuscript. The model shown here is implemented in the "development" version of ISSM, version 4.19, which is not currently publicly available. This version will be made publicly available soon, as we always do. And yes, we believe there is still room for improvement that we hope will be addressed in future work for us or anyone in the Cryosphere community.**

I have a few minor nitpicks. I try to think of any paper like this not just as a description of how the model was implemented in a particular software package but as a guide to the poor souls who might have to reimplement these ideas in whatever programming language people are using 20 or 30 years from now. The appendix shows a number of painstaking computations, and presumably getting all of these details correct is necessary to correctly implement the mathematical model that the authors have specified, which could be quite error-prone. Obviously I'm biased here but to me this speaks to the utility of software frameworks like FEniCS, dune-fem, Devito, and Firedrake, which automate away much of the symbolic derivation and generation of integration formulae. This becomes especially apparent when considering what to do in the face of a non-isothermal ice column. Any of the tools above (provided that they can work with tensor product elements) will generate the correct vertical quadrature formula to include temperature effects. The end of section 5 suggests that much of this symbolic manipulation would need to be redone to account for variable temperature. (If I've misread that then it should be reworded to make it clear that it doesn't.) In any case this more my opinion so take it with a grain of salt.

**That is a good point. We implemented the MOLHO model in an already existing code, where all the finite element method (FEM) machinery was coded based on a "classical" software architecture employed by the FEM community since the 1980s. We agree that implementing correctly all the details shown in the appendix is a difficult task, and is quite error-prone. However, we still believe that such details could help others to implement the MOLHO model in their own ice sheet models, reducing the time spent to get the formulation detailed enough to be coded. Nevertheless, we agree that new software frameworks like Fidrake and Devito bring a faster (and smoother) path for coding finite element and finite difference methods, respectively. For this type of framework, we added another appendix section (Appendix H) where we show a more compact form of the MOLHO formulation. It was based on the original work of one of the authors (Brinkerhoff and Johnson, 2015), and we hope it will help others to implement this formulation.**

I wasn't sure where the footnote at the end of section 4.3 came from regarding an iterative solver with time complexity O(n^2) came from. A well-tuned multigrid or multilevel solver for a 2D elliptic equation converges to an accuracy comparable to the discretization error much faster than this -- O(n log n) or even O(n) if you include a global coarse level. See *Domain Decomposition Methods* by Toselli and Widlund. Sparse direct solvers are asymptotically faster than O(n^2) for 2D elliptic finite element problems too.

**Actually, that was supposed to be an upper bound complexity, in the sense that the MOLHO model could demand up to 4x computational time in comparison to the SSA model considering a no tuned iterative solver (with complexity $O(n^2)$). We updated that footnote (and the main text accordingly) considering a lower bound, i.e., an optimal solver with complexity O(n).**

My final concern is that the higher-order model has a minimization principle which may not be preserved with the vertical averaging approximation that the authors use. Having a convex minimization principle means that you get more guarantees about convergence rate when you use Newton's method with a globalization strategy such as line searches or trust regions. The authors instead used Picard's method, which I've found often requires some manual and problem-specific tuning to get an acceptable convergence rate. This strikes me as more of a tradeoff than some fatal flaw and the gains in speed of the MOLHO model might outweigh any advantages of having a convex minimization principle.

Well, we do not fully understand how our vertical integration could compromise the well-posedness characteristic of the original Higher-Order model. MOLHO is an HO model with a specific shape function in the vertical, but the formulation is the same. We employed the Picard scheme because this is the default non-linear solver of ISSM (it has been used extensively for different ice sheet and glacier geometries, and it is sometimes more robust than Newton-algorithm implementations). In the experiments performed in this work, we did not observe any convergence problem with MOLHO. To illustrate how models converge during the Picard-iteration scheme, Figure 1 below shows the evolution of the residual for each Picard iteration for experiments A and C (both using the shortest wavelength, 5 km). The residual is computed as $res = || K^n u^{n-1} - F || / ||F||$ (K is the stiffness matrix, u is the solution vector, F is the loading vector, n is the Picard iteration, and $|| \; ||$ is the norm 2) and it indicates the mechanical equilibrium of the ice sheet. We employ the same stopping criteria, and we use a direct solver (MUMPS) at each Picard iteration. As seen in Figure 1, both MOLHO and HO present similar convergence rates.

[Figure]

[Figure]

Figure 1. Evolution of the mechanical equilibrium residual at each Picard iteration.

In any case I recommend for publication almost as-is and look forward to seeing future work on paleo simulations with higher order models!

Thank you again for your thorough and insightful review!

**Response to the anonymous reviewer (RC2)**

This paper presents a new vertically integrated ice flow model. This formulation approximates a classical Higher-order model (known as the Blatter-Pattyn model) by assuming a specific polynomial shape for the vertical variations of the horizontal velocities, which allows analytical integrations.

This paper gives a clear description of the model and its numerical implementation, so that it will be very useful for the community.

The model performances are tested against the original Higher-order model on standard ice flow model inter-comparison benchmarks. The results are clearly described.

**We thank the anonymous reviewer for their support and comments, which helped clarify the manuscript.**

I only have few minor comments listed below :

- The polynomial function used to approximate the horizontal velocities derives from analytical solutions of the Shallow Ice Approximation for an isothermal ice sheet, which may increase the differences in real application or will require further work. This is acknowledged in the discussion (lines 330-333). I think this should be already introduced and discussed in the derivation of the model (sec. 2.1).

  **We agree. We added a brief discussion about it in Sec. 2.1. We keep the original phrase in the Discussion Section (sec. 5).**

- The main goal of a higher order model is to approximate the Stokes solution. The discussion of the ISMIP-HOM experiments focuses on the difference between the MOLHO and HO models, it would be good to give a brief summary of the inter-comparison results and especially how HO models approximate the Stokes solution.

  **We agree. A brief overview of the ISMIP-HOM conclusions was added in the beginning of the Discussion section (Sec. 5). We also reordered the paragraphs in the Discussion such that now we discuss the ISMIP-HOM results first, and then the MISMIP3d results.**

Technical details:

- Line 48: "we precompute a depth-average "the" viscosity" => remove "the"

  **Thanks for catching it.**

- Line 127:" (e.g., v^b=[v^b_{x,1}, v^b_{x,2}, }, v^b_{x,3}])" should be "(e.g., v^b=[v^b_{x,1}, v^b_{x,2}, }, … v^b_{x,n}]) with n the number of nodes"?

  **We agree. We changed as requested. Note that we use "n_{f,e}" instead of "n" since "n_{f,e} was already defined as the number of element nodes.**

- 15 : just give a reminder that dv_z/dx and dv_y/dy are assumed to be negligible as assumed in the HO model.

**We added a reminder about it.**

- 16. An equation usually requires a comparison between terms.

**We introduce an auxiliary variable (\mu') to improve the notation of that expression.**